# Organic Management of 'Maradol' Papaya (*Carica papaya* L.) Crops: Effects on the Sensorial and Physicochemical Characteristics of Fruits

**Perla Ruiz-Coutiño, Lourdes Adriano-Anaya, Miguel Salvador-Figueroa, Didiana Gálvez-López, Raymundo Rosas-Quijano and Alfredo Vázquez-Ovando ***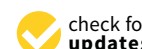

Instituto de Biociencias, Universidad Autónoma de Chiapas, Boulevard Príncipe Akishino sin Número, Colonia Solidaridad 2000, C.P. 30798 Tapachula, Chiapas, Mexico; perla_mayumi18@hotmail.com (P.R.-C.); maria.adriano@unach.mx (L.A.-A.); miguel.salvador@unach.mx (M.S.-F.); didiana.galvez@unach.mx (D.G.-L.); raymundo.rosas@unach.mx (R.R.-Q.)

**\*** Correspondence: jose.vazquez@unach.mx; Tel.: +52-962-6427972

**Abstract:** The Maradol Papaya (*Carica papaya* L.) is a fleshy berry produced in the tropics; it is highly appreciated around the world for its high nutritional and medicinal value, as well as its attractive sensory properties. Evaluating the physiological, chemical, and sensory characteristics of 'Maradol' papaya fruits from organically managed crops was the primary objective of this study. Four treatments (T1–T4) were evaluated, all of which were fertilized using the same organic management practices. In addition, plant extracts were applied regularly to T1–T3 as pest control, and single (T2) and double (T1) rows of trap plants (roselle) were used. T4 did not receive additional treatment. Fruits under conventional agriculture outside the experimental site were included for comparative purposes (controls). The organic management of the plants did not negatively influence the physiological traits of postharvest ripening. Among the organic treatments, T1 fruits had the highest total soluble solids, vitamin C, and reducing sugars, as well as the lowest weight loss, which significantly improved the quality of the fruit, compared to conventionally produced fruits. In addition, sensory evaluation performed by trained judges, revealed that fruits from the organically managed plots (T1–T4) were the softest and juiciest, and had a higher score in fruit and papaya aroma, in contrast to the conventionally produced fruits, which turned out to be sour, more astringent, and less soft and juicy. The results show that the exclusively organic management of 'Maradol' papaya crops improves several post-harvest traits of the fruits, compared to those that can be purchased commercially and are conventionally grown.

**Keywords:** organic agriculture; trained judges; biofertilizers; fruits quality

## 1. Introduction

The papaya (*Carica papaya* L.) is a plant that is native to the tropics of Mexico and Central America. Its fruit is highly appreciated around the world for its high nutritional and medicinal value, as well as its attractive sensory properties [1]. The Maradol variety stands out for its appearance, flavour, and nutritional value, which is why it is considered one of the most important cultivars [2]. Mexico occupies fifth place in the global production of this fruit [3], but it is the primary exporter to the United States' market, at more than 146,000 tons per year.

The production, quality, and safety of papaya fruits are influenced by pre and post-harvest factors [1]. The most determinant pre-harvest factors are plant fertilization and the control of pest and diseases. Conventional procedures employ chemically synthesized inputs (fertilizers and compounds for controlling pests and diseases). This strategy, which assists in plant development and

pest control, has some negative implications when overused [4]; i.e., the excessive accumulation of substances; the contamination of the soil, water, and air; and the appearance of resistance to some of the compounds [5].

Organic agriculture is a holistic production management system which promotes and enhances agro-ecosystem health, including biodiversity, biological cycles, and soil's biological activity. It emphasizes the use of waste inputs from the farm itself, considering that regional conditions require locally adapted systems. This is accomplished by using, where possible, agronomic, biological, and mechanical methods, as opposed to using synthetic materials, to fulfil any specific function within the system [6]. Some of the various strategies employed by organic agriculture include the use of organic fertilizers, plant extracts, trap plants, etc., which can be efficient alternatives that replace the effects produced by the synthetic products in conventional agriculture [7]. The use of organic compounds has less impact on the environment compared to conventional agriculture [8]. With the incorporation of organic agricultural practices, the biochemical processes of the native microbiota are accelerated, and the availability of nutrients increases, so that, for the plant, the nutrients are easier to assimilate [9].

Various types of biofertilizers (liquid bioferment, compost, vermicompost, vermicomposting leachate, and green manure), pest and disease controllers (predators, antagonistic organisms, and extracts or infusions of various plants), and trap plants (lucerne, roselle, and sunflowers, among others) can be used into organic agriculture [10,11].

The use of these products or organisms can help one to obtain greater plant health, product safety, and in some cases, characteristics of the fruits similar or better to those obtained by conventional agriculture [12]. There are examples of improvements in the physical, chemical, or sensory traits of fruits when they are treated with organic inputs, with respect to their counterparts produced with conventional agriculture. Thus, it has been documented that these products can provide sufficient nutrients for the growth of 'Grand Naine' bananas and that their use during two productive cycles promotes an increase in the total sugar and ascorbic acid contents for sweeter fruits, according to evaluations by trained judges [13]. It has also been reported that organic fertilization improves nutraceutical characteristics (specifically, the polyphenol content and the antioxidant capacity) of the cantaloupe melon [14]. The application of organic fertilizers increased the yield of cashew trees in addition to the polyphenols, fat, and titratable acidity in seeds and pseudo-fruit [10]. In addition, although there is very little evidence, it has been shown that when trap crops are used near crops of agronomic interest, the quality and/or productivity of the plants can be improved [15]. This has been shown to occur in cucurbit crops, where damage caused by predators has been reduced; the health of the plants is improved, and consequently the physicochemical and sensory quality of the fruits is increased [16]. The damage caused by stink bugs in pepper fruits was also minimized using sunflower and sorghum as trap crops [17].

Several biological products (biofertilizers, plant extracts, earthworms, and trap crops) have been evaluated for soy, banana, and corn crops, among others. For the papaya, there is information about how organic fertilization influences the yield [18,19], the soil characteristics [20], and the plant's growth [19], but there are few studies about the effects of organic management on the fruit's physicochemical quality traits [18,19], and far fewer focused on sensory characteristics. Therefore, the objective of this study was to evaluate the physicochemical and sensorial characteristics of 'Maradol' papaya fruit from organically grown plants and to compare them to fruit obtained under conventional production system.

## 2. Materials and Methods

### 2.1. Study Site and Plant Material

The experimental plots (T1–T4) were established in an orchard with exclusively organic management located 14°49′48.56′′ N, 92°17′46.98′′ W and 58 m.a.s.l. in southern Mexico (Tapachula, Chiapas). The characteristics of that location are an average annual temperature of 30.7 °C,

average annual humidity of 80%, average rainfall of 2632.9 mm (meteorological station 769043 MMTP), and loam soil (47.28% sand, 20.72% clay, and 32% silt). Before the crop was established, the weed plants were cut and left in the same place as green manure, and minimal tillage was performed to create adequate conditions for plant development.

The papaya seedlings, cultivar Maradol, were donated by AGROMOD S.A. de C.V. (Chiapas, Mexico). Roselle (*Hibiscus sabdariffa*) seeds were purchased from a local seed distributor (Hortaflor®, Querétaro, Mexico). Roselle was used as a trap crop, based on previous experiences, and because it is a plant that is moderately susceptible to predatory attack [21].

### 2.2. Treatments and Experimental Conditions

Four cultivation plots (treatments) were established using a completely randomized design over a total area of 340 m$^2$. Due to the uniformity of the experimental field in terms of physicochemical characteristics (data not shown), it was not necessary to perform the experiment in blocks. The four treatments were organically fertilized, with variations in the use or non-use of plant extracts (pest controllers), as well as the use and quantity of trap plants (Table 1). Treatments 1 and 2 were grown under the same conditions (Table 1), but they differed in the number of trap plants (T1 with a double row of roselle plants and T2 with a single row of roselle plants).

**Table 1.** Treatments established for the organically grown 'Maradol' papaya.

| Treatment | Application of Organic Fertilizers | Application of Pest Controllers (Vegetal Extracts) | Use of Trap Plants |
|:---:|:---:|:---:|:---:|
| T1 | + | + | ++ |
| T2 | + | + | + |
| T3 | + | + | − |
| T4 | + | − | − |

For the application of organic fertilizers and pest controllers, + = presence and − = absence. For the use of trap plants, ++ = double row, + = single row, and − = absence.

Each plot had an area of 67.5 m$^2$ and contained 30 papaya plants (distributed in five rows with 6 plants per row). The separation between the rows and plants was 110 cm using the "quincunx" technique, or diagonal in triangle form [22]. When the papaya plants were planted, the roselle seeds were planted 110 cm from the last furrow, forming a simple fence (T2) or a double perimeter fence (T1). A distance of 80 cm was left between the roselle plants.

### 2.3. Preparation and Application of Organic Products

Bokashi compost, liquid bioferment (called biol), and vermicompost were used as fertilizers. To manage or control pests and diseases, infusions of garlic, tobacco, wormseed, thyme, and chili tree were used, as were ethanolic extracts of cloves, chamomile, cinnamon, and garlic.

2.3.1. Preparation of Organic Products

The organic fertilizers were prepared by following the procedures described by Adriano et al. [23]. For the bokashi manure, residues from the leaves and stems of weeds, coffee pulp, sugarcane molasses, bovine manure, and lactic acid bacteria were used. The solid material was layered into a heap measuring approximately 1000 kg. Man, Rogosa, and Sharpe (MRS) broth containing lactic acid bacteria (approximately $10^{11}$ CFU mL$^{-1}$) was added at a rate of 1 L per heap. The heaps were aired manually every day for the first week, and every 7 days for the remaining five weeks.

The biol was prepared by mixing 50 kg of fresh cattle manure, 3 kg of sugarcane molasses, 4 kg of ash, 2 L of microbial inoculant to accelerate fermentation (667 mL of Sabouraud broth containing $10^9$ CFU mL$^{-1}$ of *Saccharomyces cerevisiae* + 667 mL of GYC (glucose, yeast extract and calcium carbonate) broth containing $10^{11}$ CFU mL$^{-1}$ of acetic acid bacteria and 666 mL of MRS broth containing

$10^{11}$ CFU mL$^{-1}$ of lactic acid bacteria), and drinking water to a 200 L volume. The mixture was fermented anaerobically in a hermetically sealed tank for 21 days.

The vermicompost and vermicomposting leachate were obtained through the biotransformation of coffee pulp and manure by the Californian red worm (*Eisenia foetida*). The process was performed in tanks measuring $3 \times 1 \times 1$ m while maintaining 80% humidity until the fertilizer was formed, and the leachates were collected continuously.

To prepare the vegetable infusions, we used 100 g of garlic bulbs (*Allium sativum*), 150 g of tobacco leaf powder (*Nicotiana tabacum*), 500 g of epazote (*Dysphania ambrosioides*) leaf, 150 g of leaves and flowers of thyme (*Thymus vulgaris*), and 500 g of chili tree fruit (*Capsicum annuum*). Each type of plant tissue was crushed and placed individually in containers with 9 L of boiling water. The mixture was left to stand at room temperature and stored for 14 days for later use. Another infusion was also prepared by combining 1000 g of garlic and 1000 g of chili pepper, both of which were ground in a grinder (Oskar model S0-01, Sunbeam, Miami, USA), placed in a plastic container with 10 L water and allowed to stand for 7 days.

To obtain the ethanolic extracts, 500 g of flower bud of clove (*Syzygium aromaticum*), chamomile leaves (*Matricaria chamomilla*), or cinnamon inner bark (*Cinnamomum zeylanicum*) were crushed. They were subsequently placed individually in containers with a capacity of 5 L, and 3.6 L of 70% ethanol was added. The containers were hermetically sealed and allowed to rest at room temperature for 5 days for later use. In addition, the preparation of ethanolic garlic extract was performed by reflux in Soxhlet equipment; for this purpose, 45 g was placed in a 500 mL flat bottom flask and 200 mL of 42% ethanol was added. The product of six reflows was recovered and stored at room temperature until use.

### 2.3.2. Application of Organic Products

At the time the crop was established, 2 kg of vermicompost was applied to each of the seedlings. Every 9 weeks after that (a total of five times), the same amount was applied around the stem. The biol was applied to the drip zone of the plants at sowing and every 7 days (45 times in total) thereafter at a rate of 2 L per plant. The bokashi manure was applied around the stem of the plants at a rate of 2 kg per plant every 9 weeks (five times during the experiment), alternating with the vermicompost.

The vegetable infusions were mixed as follows: 500 mL of the garlic/chili infusion + 125 mL of each of the other infusions (garlic, tobacco, epazote, thyme, and chili) + potable water to reach a final volume of 20 L. This mixture was applied by spraying the plants every 7 days (45 times) with a backpack pump and adding 15 mL of solution (for every 20 L of mixture) of neutral soap as an adherent.

The alcoholic extracts (garlic, chili, and cinnamon) were diluted individually (25 mL) to a final volume of 20 L with drinking water, and 15 mL of adherent was added. This solution was applied to crops using backpack pumps every 14 days (a total of 22 times) and alternating between each extract.

### 2.4. Crop Monitoring

During the eleven months after sowing, cultural practices such as the defoliation of papaya plants, continuous irrigation to field capacity (monitoring daily using a Kelway pH soil and moisture meter HB-2), and regular cutting and removal of weed plants were performed. During this time, the health of the fruit was visually verified, as was their correct development.

### 2.5. Sampling and Fruit Processing

Mature green fruit was harvested. The change in the peel colour from green to light green was used as the harvest index [24]. Thirty fruits randomly selected from completely healthy plants from each treatment (120 fruits in total) with similar size and shape and characteristics that were free of visual damage were cut. In addition, on the same day the organic fruits were cut, 30 fruits of the same variety, size, and same state of maturity were acquired, which were cultivated under a conventional production scheme from the Agro Pacífico S.A. of C.V. company, which is located less than 20 km

(14°54'37.8'' N 92°20'13.3'' W) from the organic experimental site. Although these fruits were grown in different fields, the climatic conditions do not change substantially between the two production sites. The conventional fruits were included for comparative purposes and to weigh up the effect of exclusively organic management, even with the limitations of using fruits of different origins.

The fruits of all the treatments (including those acquired commercially, hereinafter referred to as the controls) were washed with potable water, immersed in a 200 ppm solution of sodium hypochlorite for 3 min, and allowed to dry at room temperature. All the fruits were placed in plastic grids on tables, in a closed warehouse at room temperature (30 ± 2 °C) and 85% RH for 8 days to simulate the conditions of the humid tropics without refrigeration. From the 8th day of storage on, the fruits were transferred to a cold room (18 °C) to homogenize the ripening process, and when they reached full ripeness (six days under cold room), they were processed for instrumental and sensory analyses.

### 2.6. Physiological and Physicochemical Measurements in Fruits

From the harvest day (day 0) on, three fruits per treatment were sampled every 48 h over eight days (5 sampling dates). The physiological weight loss (WL) was monitored with a digital balance (Adventurer™ Pro model AV264C, OHAUS, Parsippany, USA). Three records were taken per fruit from the apical, middle, and peduncular areas, and the external colour and the firmness of the fruit were measured. To register the peel colour, a MiniScanEZ colorimeter was used and the *L\**, *a\**, and *b\** values of the CIELAB scale are reported. The firmness of the pulp was measured with a penetrometer (Tr®, Turoni SRL, Forli, Italy) with a conical plunger measuring 10 mm in diameter, and the results are expressed in Newtons (N). The total soluble solids (TSS) were quantified with a digital refractometer (model PAL-$\alpha$; ATAGO Co. LTD, Saitama, Japan), and the results are reported as °Brix (method 932.12) [25]. The titratable acidity (TA) was determined according to method 942.15 of the AOAC [25], and the results were reported as citric acid.

When the fruits were fully ripe, the moisture (method 934.06), protein (method 920.152; N × 6.25), ash (method 940.26), fat (method 920.39), and vitamin C (method 967.21) contents were determined according to the procedures by the AOAC [25]. Nitrogen content ($N_2$) was determined with a DK 8S heating digester and VDK152 distillation and titration unit (VELP® Scientifica, Usmate, Italy). Protein content was calculated as nitrogen × 6.25. Fat content was obtained from a 1 h hexane extraction with ST 243 Soxtec (Labtec™ FOSS, Hilleroed, Denmark). Ash content was calculated from the weight of the sample after burning at 550 °C for 2 h in a furnace model FE-350 (Felisa, Zapopan, Mexico). Moisture content was measured based on sample weight-loss after oven-drying at 110 °C for 2 h (Oven model FE-291D, Felisa®, Zapopan, Mexico). The reducing sugars (RS) content was evaluated by Miller method [26] using a spectrophotometer (Thermo Scientific Genesys 20™ Model 4001/4, Thermo Fisher Scientific Inc., Waltham, USA) and the results were expressed in mg 100 $g^{-1}$. The total polyphenols were also quantified (Folin-Ciocalteu method) [27] and the results were expressed in mg equivalents of gallic acid per g of pulp (mg EAG $g^{-1}$).

### 2.7. Sensory Evaluation

At full fruit maturity, the sensory characteristics of odour, taste, and texture were evaluated by a trained panel of judges; for that purpose, the training of the judges was performed first.

### 2.7.1. Selection and Training of Panellists

The selection and training of judges was performed by following the procedures described by Vázquez-Ovando et al. [28]. Surveys were sent to 40 undergraduate students with papaya consumption habits. The pre-selection survey was administered to assess health status, eating habits, and the ability of panellists to express the taste sensations caused by tasting a papaya. The screening criteria included good health (self-report), non-smoking, the ability to work well on a panel, interest in participating, being a consumer of papaya fruit or at least not disliking its consumption, having no allergic reactions to the consumption of papaya, and most importantly, the skills to describe and

express the sensations perceived. From the results of the survey, 14 people were selected, and they were trained over 30 sessions according to the procedures for the quantitative descriptive analysis (QDA) for taste, odour, and texture characteristics. Through group consensus, the panellists generated four odour descriptors; namely, woody, fruity, papaya, and winey. For the texture, three descriptors were generated; namely, juiciness, crunchiness and softness. For taste descriptors, the four basics, salty, sweet, bitter, and sour, were included. Unstructured scales of 15 cm were used to indicate the intensity of each descriptor. The data were subjected to an analysis of variance by panellist and by attribute, and eleven panellists (eight men and three women, with a range of ages from 19–22 years old) were chosen based on their discriminative capacity ($p < 0.30$) and repeatability ($p > 0.05$).

### 2.7.2. Sample Preparation and Evaluation

Cubes of pulp measuring 2 cm$^3$ were cut from the fruits under evaluation, taking special care not to use sections near the central cavities of the fruits. For the odour test, two portions were placed in a plastic container (110 cm$^3$) and macerated, and the container was closed and left at 25 °C for 1 h to promote the accumulation of volatiles. For the taste (sweetness, sourness, and astringency) and texture tests, ten fruit cubes were placed in a dish measuring 10 cm in diameter. Both types of containers were coded with three digits.

The final evaluation was conducted during a single session in a spacious, enclosed area with adequate lighting and a temperature of 25 °C. Coded containers were presented to the panellists. First, odour testing was performed, for which a series of five containers arranged in a circle were presented to the panellists. Each panellist was asked to remove the lid of the sealed container, inhale the volatiles and cast their judgement; to this end, each panellist was given five pages (one for each sample) on which they were asked to mark the stimulus perceived for each of the attributes (one line per attribute). This procedure was conducted two more times (in triplicate) on an unstructured, linear scale of 150 mm (transformed to a score of 0–10), using codes consisting of three different digits for each sample and repetition. Similarly, the panellists were asked to evaluate the samples for their taste and texture attributes [29]. For these evaluations, the panellists were asked to rinse their mouths with water between samples. When required, the panellists used tea bags to desaturate their olfactory system or flavourless biscuits to desaturate the taste.

### 2.8. Data Analysis

All the resulting data were subjected to an analysis of variance (ANOVA) and subsequent comparison of means using Tukey's test with Statgraphics Centurion XV software version 15.2.06 (Statgraphics Technologies, Inc., The Plains, USA). When samples were taken during ripening, comparisons were made per variable and per day, between treatments.

## 3. Results and Discussion

### 3.1. Physicochemical Characteristics of Fruits during Ripening

In general, the application of the different organic products in addition to the use of trap plants during the cultivation of 'Maradol' papayas did not negatively modify the physiological traits of their post-harvest ripening process. On the contrary, some physicochemical traits showed that the shelf life of the fruits of the T1–T4 treatments was prolonged compared to the fruits obtained from a plot with conventional agriculture.

Regarding weight loss, significant differences between treatments were observed from the second day of storage on. The fruits grown in the conventional system (controls) presented the greatest physiological weight loss (9.62%) out of all the treatments, on the 8th day of storage at room temperature (Figure 1A). This same behaviour was observed at the end of the refrigerated storage period, because on the day of consumption, the control fruits presented the greatest weight loss. As a result of this weight loss, fruits with lower moisture contents were obtained (79.40%). Fruits from plants grown

with biofertilizers + extracts and infusions + one (T2) and two fences of trap plants (T1) showed the lowest weight loss compared to the other treatments throughout the storage, with the difference being significantly different ($p < 0.05$) regarding controls. These treatments, in addition to organic fertilization, had in common that they were surrounded by trap crops. The above demonstrates a protection effect or roselle, as has been shown to occur with other trap crops [17] and this helped reduce the papayas' weight loss. Similar results were reported by Reganold et al. [30], who found that weight loss in strawberries was significantly lower in organic crops (25.40%) compared to those grown conventionally (27.52%). In papayas, applying 25% bio-compost to a conventional culture system reduced the weight loss by 11.20% compared to an exclusively conventional treatment [18]. These results suggest that organic papayas would have a longer shelf life than conventional papayas as a result of the decrease in dehydration and decomposition, possibly due to the increase in the cuticle and the walls of epidermal cells [30]. The above was a result of the increase in the biosynthesis of fatty acids components of the cuticle promoted by plant hormones produced during the organic management [31]. Another explanation is that these structural and other biochemical changes help to minimize the respiratory stress that fruits suffer during the postharvest period.

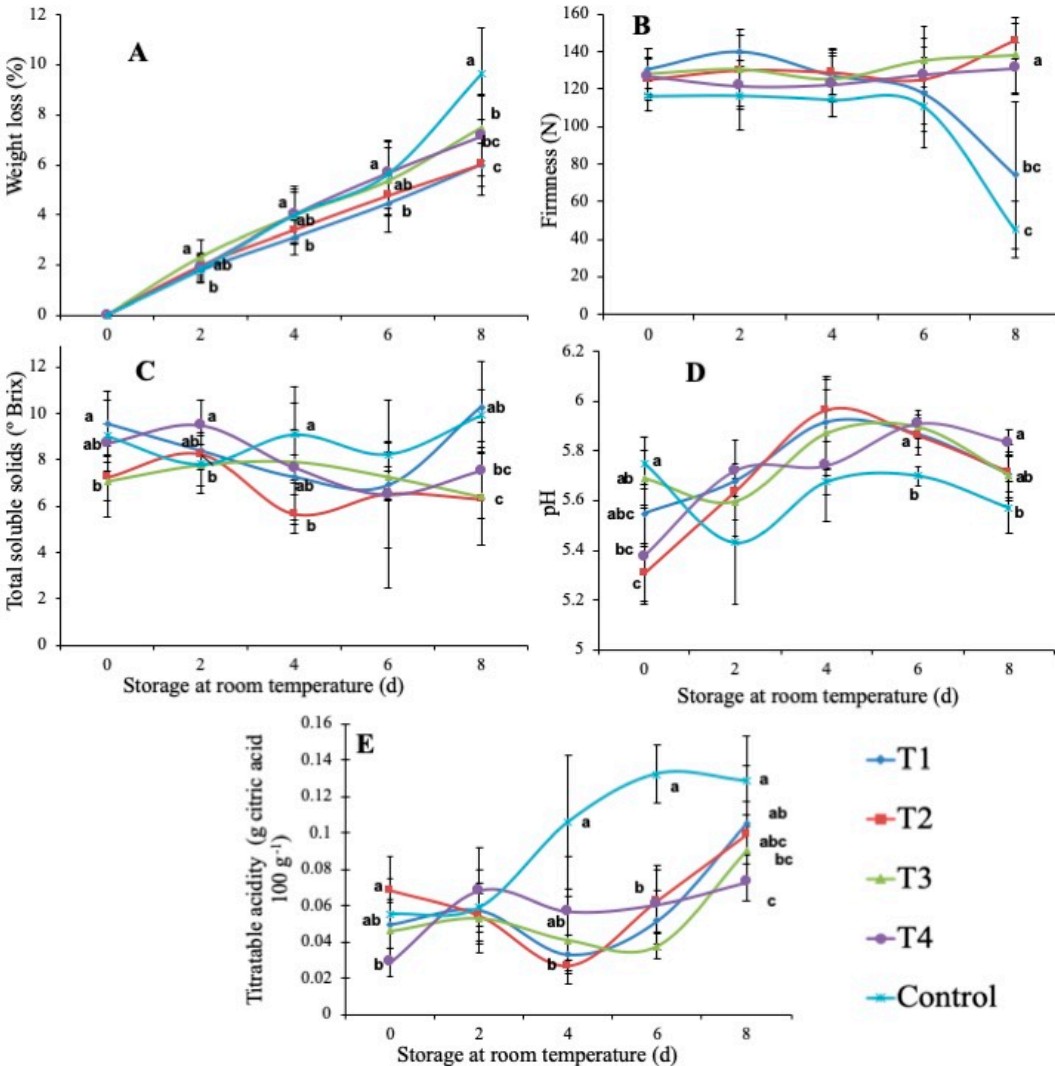

**Figure 1.** Weight loss (**A**), firmness (**B**), total soluble solids (**C**), pH (**D**), and titratable acidity (**E**) of 'Maradol' papaya fruits from plants grown under different production schemes and stored at $30 \pm 2$ °C and 80% RH. Different letters (per day) denoted significantly differences ($p < 0.05$). For treatment details, see Table 1.

However, the fruits from the T2, T3, and T4 treatments retained greater firmness during the storage days (Figure 1B). The differences for this trait were significant ($p < 0.05$) on day 8, compared with T1 and control treatments. From the sixth day of storage on, these treatments showed a dramatic decrease in the firmness of the pulp, with values below 80 N. Xiang et al. [19] reported that papaya plants treated with compost had firmer fruits, which is highly desirable because they represent fewer complications during storage. The primary factor responsible for the loss of firmness is the enzymatic degradation of the cell wall components [32], and indirectly, the weight loss, primarily due to the loss of water, and consequently, of cellular turgidity. The foregoing results were not fulfilled completely according to the observations of the T1 fruits, which presented low weight loss and a pronounced loss of firmness.

The TSS content of Maradol papaya fruits showed significant differences ($p < 0.05$) between treatments (Figure 1C). The T1, T4, and controls presented the highest values for this trait. Xiang et al. [19] reported differences in the TSS contents among the papaya fruits of plants treated with compost, which presented 12.96% more TSS than untreated plants. Sarhan et al. [33] demonstrated a similar effect on *Cucurbita pepo* L. when using organic fertilizers (sheep manure) and bacteria from the genus *Azotobacter*; both treatments also increased the fruit production. During storage, there was no increase in the TSS content, and, by contrast, treatments T2–T4 showed a decrease in the TSS (Figure 1C). Even with this result, the variation in the TSS content during the postharvest period tends to be minimal, and the values are similar to those of other reports. Gomez et al. [34] reported values of 9.5% TSS in mature green papaya fruits and 10% in fully ripe fruits.

By contrast, the pH and TA showed significant differences ($p < 0.05$) between treatments for most days of the evaluation (Figure 1D,E, respectively). The fruits from the control treatment were different on most of the days of the remaining treatments, generally presenting the lowest pH values (Figure 1D) and the highest TA values (Figure 1E). Towards the end of storage (day 8), the fruits showed a slight decrease in the pH and TA. According to Gómez et al. [34], there are several studies reporting that the TA tends to increase as the fruits ripen completely.

The peel colour is the first visual attribute that is evaluated in papaya fruits. Table 2 shows the values of the *L\* a\** and *b\** traits that are indicative of the fruit colour. At the beginning of the study (day zero), the values were slight differences between treatments and the differences can be explained by the variability between the fruits. However, during days 2, 4, and 6, the green fruits had similar *L\** and *b\** values between treatments, which is because these traits undergo fewer changes during papaya ripening. The *a\** value changed dramatically during storage and showed significant differences between treatments for days 2, 4, 6, and 8. The negative values of this variable are associated with a greenish hue, and as ripening progressed, there was a tendency towards positive values associated with the reddish hue, giving the characteristic colour of ripe papaya. On day 8 of storage, the fruits from the T1–T3 treatments were the least mature from a colouring perspective. The above demonstrates the effect of infusions of plants used as pest controllers on the *a\** value, as it is common among these treatments. A very similar behaviour with minor variations between treatments was found for the *b\**. At the end of storage, in most treatments, the *L\** and *b\** values, the colour results for the Maradol papayas, were next to those previously reported, at *L\** = 55 and *b\** = 50 [24].

**Table 2.** Peel colour of 'Maradol' papayas stored at 30 ± 2 °C and an RH of 80 ± 5%. *L*\* (lightness), *a*\* (hue angle green–red), and *b*\* (hue angle blue–yellow).

| Treatments | Storage Time (Days) | | | | |
|---|---|---|---|---|---|
| | 0 | 2 | 4 | 6 | 8 |
| | *L*\*Value | | | | |
| T1 | 47.17 a | 53.54 a | 56.63 a | 58.16 a | 55.69 a |
| T2 | 42.84 c | 45.62 a | 48.74 c | 51.84 b | 53.23 b |
| T3 | 44.87 b | 49.21 a | 50.72 b | 51.74 b | 51.98 c |
| T4 | 47.99 a | 48.64 a | 51.86 b | 55.93 a | 57.09 a |
| Control | 46.98 a | 50.14 a | 49.64 bc | 57.66 a | 57.06 ab |
| | *a*\*Value | | | | |
| T1 | −8.75 b | −1.90 a | −4.64 a | −4.31 b | −3.58 b |
| T2 | −9.20 bc | −8.14 b | −8.03 c | −4.06 b | −2.25 b |
| T3 | −9.02 bc | −8.70 b | −6.83 bc | −4.51 b | −5.90 c |
| T4 | −9.42 c | −7.52 b | −8.29 c | −5.45 b | −0.29 a |
| Control | −5.06 a | −3.10 a | −5.70 ab | −4.41 a | 0.00 b |
| | *b*\*Value | | | | |
| T1 | 29.53 a | 51.65 a | 56.84 a | 41.21 a | 42.72 b |
| T2 | 24.88 c | 31.84 a | 31.90 a | 37.52 a | 39.91 b |
| T3 | 26.76 b | 32.47 a | 31.78 a | 36.76 a | 43.65 b |
| T4 | 29.86 a | 36.16 a | 33.30 a | 38.09 ab | 50.35 a |
| Control | 26.04 bc | 39.72 a | 43.27 a | 35.83 a | 39.73 b |

Values followed by different letters (per column) are significantly different ($p < 0.05$). For treatments details, see Table 1.

### 3.2. Characteristics of Fully Ripe Fruits

Table 3 summarizes the characteristics measured for the fruits when they were fully ripe, a state in which the effects caused by the different fertilizer applications on the physicochemical traits and the quality characteristics of the fruit are better appreciated. The agronomic activities performed in the T1 treatment significantly improved the quality of the fruit with respect to controls, since the former had the highest TSS and ascorbic acid values and the lowest weight loss. With the exception of T1, the other treatments showed slightly low values in terms of TSS content compared to that reported for Maradol papaya fruits when consumed at maturity; their values ranged from 10–11.5% [24]. Other studies have reported the beneficial effect of using organic products on the TSS in papayas [19] and tomatoes [35], on the RS contents of papayas [18] and bananas [13], and on the vitamin C quantities in tomatoes [35] and bananas [13]. The strongest reason for these effects seems to be the increase in the microbial activity of the soil promoted by the bio-products, which translates into the greater availability and subsequent transfer of nutrients to the plants, possibly more effectively than what happens in conventional fertilization.

The protein and fat contents did not show significant differences ($p > 0.05$) between treatments, which may be due to the low amounts of those biomolecules that accumulate in papayas, since in other oilseed crops, there was a positive effect from the addition of organic products on the accumulation of higher fat contents [10]. The ash content was higher in treatments T3 and T4, while the lowest value was found in the fruits of T2. Since all the fruits were grown in the same place and fertilized in the same way, the only explanation for the differences is that the plants used as traps in T1 and T2 were able to assimilate part of the nutrients that were applied to papaya plants. Although a beneficial effect of organic fertilization on the increase in polyphenol content is reported in many crops [10,36], in the present study, no differences were found in the contents of these compounds, even if, as in some organic management treatments, the content was numerically lower than that of the control. In grapes [37] and yellow plums [38], a decrease in the polyphenol content was reported when plants were organically

fertilized. It is possible that there is an increase in the content of polyphenol in organically grown plants, but at the same time, these compounds are used for the synthesis of other polymers that act as defence mechanisms [36].

**Table 3.** Physical characteristics and chemical composition of Maradol papaya fruits grown using different agronomic strategies and stored for 8 days at 30 ± 2 °C, with the subsequent 7 days at 15 °C.

| Parameter | | T1 | T2 | T3 | T4 | Control |
|---|---|---|---|---|---|---|
| Moisture (%) | | 88.32 ± 1.70 ab | 83.12 ± 5.98 bc | 91.02 ± 1.68 a | 88.47 ± 4.41 ab | 79.40 ± 6.54 c |
| Protein (g 100 g$^{-1}$) | | 0.50 ± 0.14 a | 0.55 ± 0.15 a | 0.60 ± 0.08 a | 0.56 ± 0.20 a | 0.45 ± 0.07 a |
| Fat (g 100 g$^{-1}$) | | 0.08 ± 0.06 a | 0.16 ± 0.07 a | 0.12 ± 0.10 a | 0.11 ± 0.09 a | 0.10 ± 0.09 a |
| Ash (g 100 g$^{-1}$) | | 0.86 ± 0.17 ab | 0.34 ± 0.07 c | 1.20 ± 0.49 a | 1.23 ± 0.28 a | 0.67 ± 0.20 bc |
| RS (g 100 g$^{-1}$) | | 3.2 ± 0.2 a | 2.5 ± 0.4 bc | 3.0 ± 0.3 ab | 2.4 ± 0.4 c | 2.7 ± 0.4 abc |
| Vitamin C (mg ascorbic acid 100 g$^{-1}$) | | 44.10 ± 2.90 a | 33.78 ± 5.90 ab | 32.44 ± 3.67 ab | 31.50 ± 3.87 b | 31.78 ± 4.77 b |
| TSS (°Brix) | | 10.07 ± 0.80 a | 8.23 ± 0.32 c | 9.40 ± 0.58 ab | 9.09 ± 0.44 bc | 8.81 ± 0.86 bc |
| TA (g citric acid 100 g$^{-1}$) | | 0.13 ± 0.03 b | 0.11 ± 0.02 b | 0.13 ± 0.02 b | 0.19 ± 0.05 a | 0.11 ± 0.02 b |
| pH | | 5.46 ± 0.14 ab | 5.58 ± 0.07 ab | 5.62 ± 0.14 a | 5.27 ± 0.27 b | 5.59 ± 0.13 a |
| Total polyphenols (mg GAE g$^{-1}$) | | 1.97±0.25 a | 1.81±0.21 a | 1.88±0.12 a | 2.15±0.47 a | 2.15±0.19 a |
| Weight loss (%) | | 8.03 ± 2.08 b | 8.03 ± 2.10 b | 10.79 ± 3.41 ab | 10.76 ± 3.16 ab | 15.61 ± 0.43 a |
| Firmness (N) | | 64.86 ± 5.5 c | 106.12 ± 6.7 a | 87.47 ± 6.68 b | 68.05 ± 5.05 c | 30.87 ± 4.78 d |
| | *L** | 55.32 a | 57.14 a | 54.54 ab | 57.95 a | 50.34 b |
| Colour | *a** | 24.69 a | 21.76 a | 17.99 a | 25.44 a | 28.20 a |
| | *b** | 47.03 a | 48.23 a | 47.40 a | 50.04 a | 39.57 a |

Values followed by different letters (per row) are significantly different ($p < 0.05$). For treatments details, see Table 1. Protein = N × 6.25; RS = reducing sugars; TA = titratable acidity; and GAE = gallic acid equivalent.

### 3.3. Sensory Characteristics of 'Maradol' Papayas

The results obtained from the sensory evaluation on the descriptors of taste (Figure 2A), odour (Figure 2B), and texture (Figure 2C) for the 'Maradol' papayas are shown in Figure 2. At least one treatment was significantly different from the rest ($p < 0.05$) for all attributes evaluated by trained participants. Regarding the taste descriptors, sweetness had much higher average scores than sourness and astringency (Figure 2A). This is because the primary sugars present in papaya fruits, such as sucrose, glucose, and fructose [39], are found in quantities greater than the molecules responsible for sourness or astringency (Table 3), sensations (especially sourness) that are, therefore, "masked" or diluted by the perception of sweetness (Gómez et al., 2002). The fruits from the T1 treatment were the sweetest (9.62) and their rating was different ($p < 0.05$) from the treatments T2, T4, and control. This result confirms the chemical composition findings (Table 3), since the RS and TSS contents found in fruits from plants grown with organic fertilizers were higher than those found in fruits from conventionally fertilized plants. This correlation between the sweetness and TSS content has also been reported in strawberries [30] and 'Grand Naine' bananas [13] from organically grown plants. In general, the sweetest fruits (T1–T3), in addition to being organically fertilized, were treated with infusions of plants. Possibly, the application of the extracts enhances this characteristic, since it has been shown before that organic fertilization alone increases the sweetness of the fruits [30].

The opposite behaviour occurred for the sour taste, in which the judges gave the fruits from the control treatment the highest value (Figure 2A), with this treatment being different from the others ($p < 0.05$). The values granted by the judges are similar to those reported by Gómez et al. [34], who reported scores of 7.48 for the sweetness and 0.58 for the sourness of 'Solo' papayas grown in Brazil; but are different from those reported by Sesma-Morales et al. [40], who reported sweetness = 3.65, sourness = 0.78, and astringency = 0.65 for 'Maradol' papayas. A similar behaviour was found for astringency. The highest value was obtained for the conventional treatment, with this value being significantly different ($p < 0.05$) from the remaining four treatments. This result coincides with the polyphenolic compound content, since this same treatment presented the highest content (Table 3) of these molecules, and, although other molecules such as amino acid residues can contribute to this perception [41], the polyphenols are primarily responsible for the astringent taste in the fruits.

Although, as previously mentioned, astringency can be masked by sweetness, high scores for this descriptor are undesirable in ripe fruits that are consumed as desserts.

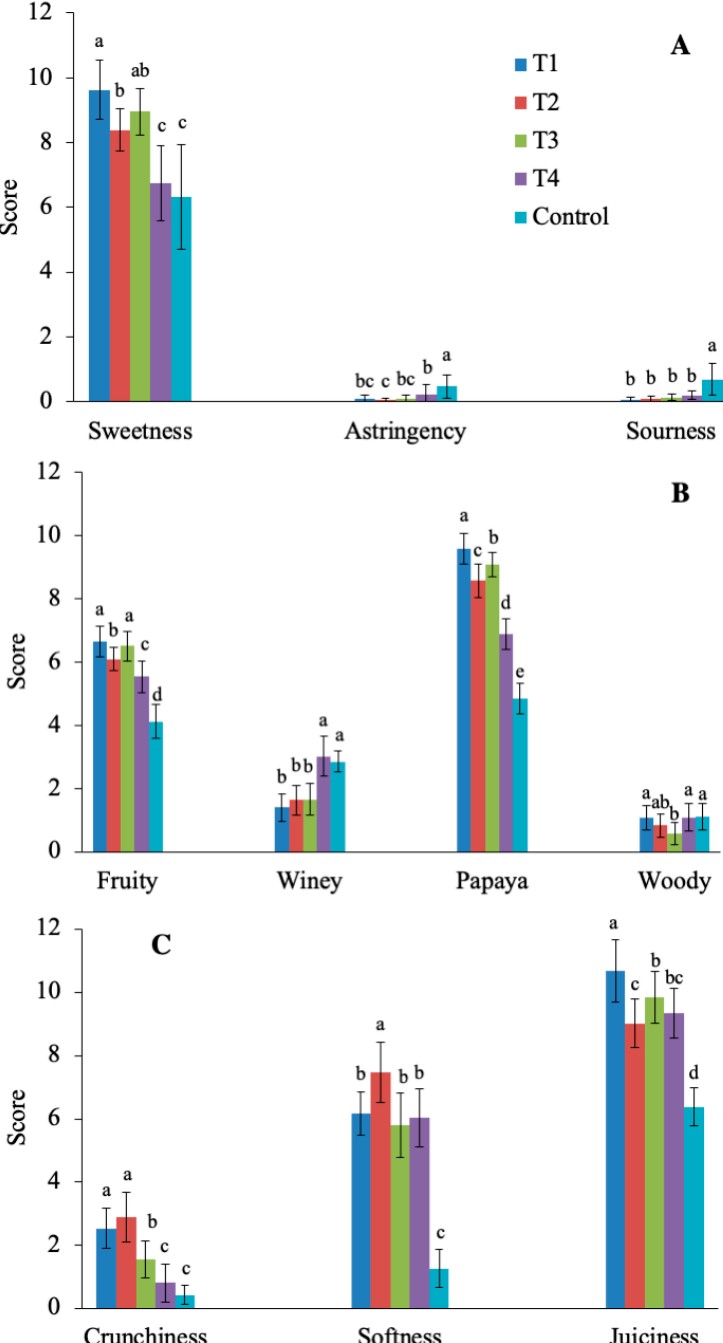

**Figure 2.** Sensory evaluation of the attributes of taste (**A**), odour (**B**), and texture (**C**), as evaluated in 'Maradol' papayas from plants grown under different production schemes and stored at 30 ± 2 °C and 80% RH. The bars represent the average values from eight trained judges. The lines on the bars indicate the standard deviations. A common letter per attribute denotes statistical equality (Tukey's test, $p > 0.05$). T1 = blue bar, T2 = red bar, T3 = green bar, T4 = purple bar, and control = blue bar. For the details of the treatments, see Table 1.

Fruity and papaya-specific odours were predominant in the samples, while winey and woody ratings had low values for all the treatments. This could have a positive impact on consumer acceptance. For all the descriptors, significant differences ($p < 0.05$) were found between treatments (Figure 2B).

The judges gave the highest values for fruity and papaya odours to the T1 treatment. Pino [42] mentioned that the most abundant aromatic component in Maradol variety papaya fruits is methyl butanoate, a compound that is also abundant in other tropical fruits [43], and this compound contributes a great deal to the typical flavour of the fruit. For the papaya and fruity odours, the judges granted the lowest values to the samples from the control treatment. Reports have shown that both a deficiency [44] and an excess of nitrogen [45] during fertilization will reduce the intensity of sensory attributes, and they tend to increase the TSS and TA contents in tomatoes [45]. Fruits from plants treated under the organic system probably received a better nutrient balance compared to conventionally fertilized plants. The increase in the volatile concentration characteristic of fruit aromas has been reported as a consequence of the use of compost, and the concentration of these compounds is proposed as an indicator of the organic management of nectarine fruits [46].

In regard to the texture descriptors of papaya pulp, the judges awarded the highest grades to juiciness and softness, with crunchiness receiving the lowest values (Figure 2C). Significant differences ($p < 0.05$) were found in the three descriptors for at least one treatment, and in general, the fruits from the control treatment always had the lowest values, although for crunchiness the control was not significantly different ($p > 0.05$) to T4. The juiciness of the papayas seems to be directly associated with the softness detected by the judges, since the fruits from the juiciest treatments were rated as less firm (Figure 1B). Contrary to the results of the present study, it has been reported that organic and integrated treatments of kiwifruit plants resulted in lower juiciness and greater fruit crunchiness compared to conventionally grown fruit, showing less general acceptability for organic fruit in consumer tests [47]. The nature and physiology of the plant may be the explanation for this contrast, since it has been reported that in annual plants, such as papayas, the effect of organic management can be faster and more efficient [10]. Higher softness values are desirable because they facilitate the chewing and the release of sugars, volatiles, and other molecules responsible for sensory attributes [34]. The control treatment showed an unusual pattern of softness, since fruits with less instrumental firmness are expected to have greater sensory smoothness, something that did not occur in this study and that is difficult to explain from a theoretical perspective.

## 4. Conclusions

In this study, applying organic compounds to Maradol papaya crops and using roselle as trap crops without any chemically-synthesized products, improved the quality of the harvested fruits compared to the conventional treatment. These treatments led to fruits with higher contents of ash (treatments T3 and T4), TSS (T1), and vitamin C (T1), in addition to sensory characteristics, indicating that the organic fruits were sweeter (T1–T3), less sour (T1–T4), softer (T1–T4), juicier (T1–T4), and had a stronger papaya aroma (T1–T4) than conventionally-grown fruits. These results suggest that the organic agronomic management of Maradol papaya plants can increase the physicochemical quality of the fruits during the postharvest, as well as the sensory quality of full ripe fruits. In addition, suppressing the use of synthetic compounds increases the sustainability of the culturing system.

**Author Contributions:** L.A.-A., M.S.-F., and A.V.-O. conceived and design the study; P.R.-C. and L.A.-A. performed the experiments; L.A.-A., M.S.-F., and A.V.-O. provided the materials and resources; P.R.-C., R.R.-Q., and D.G.-L. analysed the data; P.R.-C. and A.V.-O. wrote the paper; all authors revised the manuscript.

**Funding:** This research received no external funding.

**Conflicts of Interest:** The authors declare no conflict of interest.

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
