# Peer review of "Organic Management of ‘Maradol’ Papaya (Carica papaya L.) Crops: Effects on the Sensorial and Physicochemical Characteristics of Fruits"

_agriculture, doi:10.3390/agriculture9110234_

Round 1
Reviewer 1 Report
The authors have addressed all the comments.
Author Response
We thank the reviewer for reviewing the R1 version of the manuscript and for their opinion.
Reviewer 2 Report
I have taken note of the improvements made by the authors of the article.
However, I still have a strong reservation about comparing conventional fruits from other sites with organic fruits from the same experimental site. In my opinion, there is a significant risk of confusion of effects. I take note of the authors' response regarding the fact that conventional fruits were acquired nearby. I also noted that from a regulatory point of view, conventional cultivation on the same site was not possible, even if, given the scope of the study, I am surprised, however, that an experimental derogation was not requested. In any case, it seems to me that the scope of the results is therefore limited and that the authors must report on this limit in the manuscript.
I also note that the results do not allow to identify a single trend and this should be further highlighted and discussed in the paper. If T1 is significantly different from Control for several variables, it is not systematically significantly different from other treatments. The summary should therefore be amended.
Moreover, the differences or lack of differences between T1 to T4 treatments should be much more discussed since it seems to me that the heart of experimental design is there.
Specific comments:
L24 : why is T1 described as exclusively organic ? are the other treatments not exclusively organic ?
L46 : why should management practices be specific of organic farming ? is there a word missing ?
L74 : to which quality do you refer here ? cosmetic quality ? physio-chemical quality ? sensory quality ? the underlying assumptions about the effect of agricultural practices are not at all the same according to the type of quality studied, so I think there is a need to be more specific
L306: is this not inconsistent with the assumption made in line 404 stating that the organic management of the plants may have been efficient against the attack of pests and diseases ?
L396: greater availability compared to what kind of fertilization management ? when chemical fertilizers are used, it seems to me that the objective is that the nutrients are immediately available for the plants and in sufficient quantities...
Round 2
Reviewer 2 Report
Even if some aspects could have been further discussed from my point of view (see previous report), the authors have significantly improved the manuscript during successive reviews.
Author Response
Dear Reviewer;
In the new version of the manuscript we have incorporated several sentences that increase the discussion of the results among organic management treatments.
Thank you for your review.
This manuscript is a resubmission of an earlier submission. The following is a list of the peer review reports and author responses from that submission.
Round 1
Reviewer 1 Report
Please find my suggestions and a few questions on the attached manuscript for the authors to attend to. Thank you.

Reviewer 2 Report
Comments on the whole article:
This manuscript describes the effect of different organic management on papaya fruit quality. Contrasting results have been published regarding the effects of organic farming on product quality. The topic is therefore of interest to the readership of agriculture. In addition, the manuscript studies both physiochemical and sensorial quality, which is in my view valuable. However, I have some significant concerns regarding the design of the study and regarding the redaction of the paper.
First, organic fruits acquired in experimental conditions were compared to purchased conventional fruits. The purchased fruits might have been grown in microclimatic conditions (e.g. temperature, solar radiation, humidity…) or soil conditions quite different from the fruits harvested in the experimental treatments. Tomato chemical composition is for instance quite sensitive to environmental factors. How can you ensure that the measured differences between harvested fruits from the experimental treatments and acquired fruits are related to production factors and not to environmental factors ? In my view, if no precise information are available regarding the environmental conditions of the acquired fruits, no conclusion regarding the effect of conventional production vs. organic production should be drawn
My second concern regards the conclusion. Although I have misunderstood something, I think that the conclusion is not consistent with the results. Not all treatments lead to values significantly different from conventional ones regarding the studied characteristics and TSS and vitamin C in particular. I therefore don’t understand how the authors can conclude that applying organic compounds to Maradol papaya crops improved the quality of the fruits compared to the conventional treatment. In addition, as mentioned hereabove, as fruits of the control treatment were not grown on the same site, other factors than production factors might explain the observed differences.
My last concern deals with the introduction. There is a focus L49-56 on fertilizer effects on fruit quality whereas the paper deals afterwards with the effects of four treatments based on the use or no-use of plant extracts as well as on the use and quantity of trap plants. As far as I am aware, there are no difference regarding fertilization between the four treatments. I have therefore the feeling that the introduction is not consistent with the following parts, making the paper difficult to read.
Specific comments:
L.40 : please refer to a broader definition of organic farming
L41 : ‘that minimize the use of in the best cases, replaces the use of synthetic products’ -> are they not forbidden in organic farming ?
L42-43 : ‘conventional and intensive agriculture’ do you refer to the same type of agriculture or to two types ? please rephrase -> compared to synthetic ones
L43 : please add a reference
L46 :please explain already here (and not only L.88) what ‘biol’ are ?
L47 : pest and disease controllers can include other items than extracts or infusions of various plants. Please rephrase
L48 : ‘can be incorporated into agricultural crops’ how can biofertilizers be incorporated into crop ? Please rephrase
L49 : this sentence is in my view unclear and is problematic because making reference to very different type of products. According to the available literature, biofertilizers or pest and disease controllers for instance affect fruit quality through different mecanisms and different components of fruit quality may be affected. Please rephrase.
L51 ‘an increase’, L53 : improves nutraceutical characteristics’, L55 : ‘increases the yield’. Please specify what the baseline or reference is (for instance : no fertilisation, synthetic fertilisation…)
L57 : are ‘biological products’ referring to fertilisers ? please be more specific
L60 : are sensory characteristics not part of fruit quality parameters ? if not, please specify what quality paratemeters are.
L138 : how was irrigation at field capacity motinored ?
L.139-140 : how were fruit health and development monitored ?
L.142 : why were fruits harvested only from completly healthy plants ? were numerous plants not completly healthy ?
L.144 : please explain why only fruits with similar size, shape and characteristics were sampled.
L.145-147 : surely fruits were not acquired and then cultivated. Please rephrase. Do you mean that 30 fruits of the same variety and same state of maturity that had been cultivated under a traditional production scheme were acquired ? In addition, please be more specific regarding the traditional production scheme : do you refer to conventional production ?
L.149 : please explain why fruits were immersed in a 200 ppm solution of sodium hypochlorite. Is that common in papaya production ?
L.172-174 : misplaced, please suppress
L.180-181 : please give minima insights on the survey performed and on the selection criteria
L.231 : could you make hypothesis explaining the possible increase in the cuticle and the walls of epidermal cells in relation with the applied treatments?
L.286 : if I am reading the table correctly and as common letters are shared with other treatments, T1 TSS values are not significantly different from all other treatments and are only significantly different from T2, T4 and control values. Please rephrase or justify why T1 TSS value are significantly different from all other values
L.317 : similar comment as L.286
L.350-351 : yes, but there are also significant differences among treatments. How would you explain such a result?
L.360-361 : except for crunchiness
